

# Detection of partial discharge patterns in hybrid high voltage power transmission lines based on parallel recognition method

Yang Yang[1], Yongye Wu[2], Yifei Gao[3], Yixuan Huang[4], Shukun Liu[5] and Yuanshi Wang[6]

[1] Xihua University, Chengdu, China
[2] Chengdu Power Supply Company, State Grid Sichuan Electric Power Company, Chengdu, China
[3] Chongqing University of Posts and Telecommunications, Chongqing, China
[4] Queen Mary University of London, London, United Kingdom
[5] Nanyang Technological University, Nanyang, Singapore
[6] University of Sheffield, Sheffield, United Kingdom

Corresponding author
Yang Yang,
yangyang4@stu.xhu.edu.cn

## ABSTRACT

Due to their specially designed structures, the partial discharge detection of hybrid high-voltage power transmission lines (HHVPTL) composed of overhead lines and power cables has made it difficult to monitor the conditions of power transmission lines. A parallel recognition method for partial discharge patterns of HHVPTLs is proposed by implementing wavelet analysis and improved backpropagation neural network (BPNN) to address the shortcomings of low efficiency, poor accuracy, and inability to parallel analysis of current partial discharge (PD) detection algorithms for HHVPTLs. Firstly, considering the non-smoothness of the partial discharge of the HHVPTLs, the wavelet packet decomposition algorithm is implemented to decompose the PD of the HHVPTL and resolve the relevant signal indicators to form the attribute vectors. Then, BPNN is implemented as a classification model. A beetle optimization (DBO) algorithm based on orthogonal contrastive learning improvement is introduced to optimize the BPNN parameters since BPNN has a slow convergence problem and fails easily into a local optimum. The proposed IDBO-BPNN is employed as the model that recognizes and analyzes the parallel partial discharge patterns of HHVPTLs. Finally, the suggested model is implemented to investigate the local discharge data of an HHVPTL in the Kaggle Featured Prediction Competition and is compared with other algorithms. The experimental results indicate that the proposed model can more accurately identify whether PDs occur in an HHVPTL and detect phases where PDs occur, with higher overall accuracy and efficiency. An excellent practical performance is achieved. The proposed model can achieve the recognition accuracy of 95.5%, which is 5.3333% higher than that of the DBO-BPNN and far more than other recognition algorithms.

# INTRODUCTION

Hybrid high voltage power transmission lines (HHVPTL) composed of overhead lines and power cables are extremely substantial components in the power system and currently belong to the increasingly complex operating environment of the power system. HHVPTL gradually becomes the main artery of the power system (*Venkatesh et al., 2020*; *Ding & Zhu, 2021*). HHVPTL are prone to sudden failures due to long-term high voltage and high-current operation, resulting in external damage, equipment defects, and damaged insulation (*Ragusa, Sasse & Duffy, 2021*). Partial discharge (PD) monitoring is an efficient means to evaluate the insulation status of HHVTPLs and to assure their safe and stable operations (*Houdai et al., 2022*; *Yamashita et al., 2018*). The detection of the local discharge pattern of HHVTPLs is one of the important aspects of monitoring and diagnosis. However, due to the extremely complex compositional structure of HHVTPLs, the accuracy of PD diagnosis carried out manually is insufficient to meet the needs of power grid engineering. Therefore, pattern recognition methods with higher accuracy are needed.

The rest of the article is outlined as follows: the 'Literature Review' section is allocated to the literature review, Section 'Feature Analysis and Extraction of the PD Signal for Hybrid High-Voltage Power Transmission Lines' deals with extracting and assessing PDs' features that can be used in the proposed algorithm, The proposed method and its fundamental notions are presented in Section 'IDBO-BPNN-Based Pattern Recognition Model For Local Discharge Signals'. The experiments and comparisons are conducted in Section 'Experimental Test'. The 'Discussion' section is allocated to the discussion. The research is concluded in the 'Conclusion'.

# LITERATURE REVIEW

To resolve the previously mentioned difficulties, more advanced emerging technologies are being applied to recognize the PD pattern of HHVTPLs. Currently, there are two main methods to recognize the local discharge pattern of power equipment. The first group is based on the manual identification of the collected PD data by experts, but the results are subjective. The other one is to train the local discharge pattern recognition model by implementing machine learning methods. *Cheng et al. (2019)* obtained the charge characteristics of cable joints by applying different levels of voltage and accomplished efficient identification of cable defects based on vector quantization theory and fast-matching algorithms. *Zhang et al. (2022)* proposed a novel approach to detect the PD pattern of power transformers. The ultrasonic signals caused by PD through the Fabry–Perot (F-P) ultrasonic probe sensing array are collected, and then the time-frequency representation is employed to convert the ultrasonic signals into a time-frequency matrix. Finally, the recognition of PD is performed through a modified ResNet-18 network. *Huang et al. (2022)* implemented an ultra-high frequency (UHF) sensor to acquire the phase-resolved partial discharge (PRPD) and phase resolved pulse sequence (PRPS) spectra and then designed a conjoined fusion network to accomplish the pattern recognition of PD. *Chang, Chang & Boyanapalli (2022)* designed a probabilistic safety assessment (PSA) method to identify the insulation status of power cable joints with PD by implementing a

convolutional neural network (CNN) algorithm without measuring voltage signals. *Yao et al. (2020)* suggested that the data features to recognize the PD pattern of power equipment have high dimensionality and may contain redundant information. Thus, the random forest variance analysis (RF-VA) method was proposed to screen the optimal subset, which effectively reduced the dimensionality of the PD attribution set and enhanced the accuracy of PD defect detection.

However, the research on PD pattern recognition of HHVPTLs has been still relatively limited so far since the local discharge signal has uncertainty led by the special characteristics of HHVPTLs, which is primarily represented by the concentration of the main frequency and the poor effect of frequency analysis. Considering the actual environment of current power grid projects, PD pattern recognition is highly required for HHVPTLs. Therefore, the article proposes wavelet packet decomposition to analyze the time-frequency decomposition of PD signals. Moreover, the characteristics of the PD signal of HHVPTLs are deeply analyzed. So, a PD feature vector is built on this basis. Then, the improved backpropagation neural network (BPNN) utilizes them as inputs to train the algorithm. Eventually, a model that recognizes the PD pattern is obtained. Thus, the effective recognition of PD of HHVPTLs is realized, and the PD recognition technology of HHVPTLs is promoted. More up-to-date research presenting some practical modifications can be found in *Faizol et al. (2023)*, *Suren & Parvatha (2013)*, *Hussain et al. (2023)* and *Shanmugam et al. (2023)*.

The highlights of the article are:

1. Wavelet packet decomposition is employed to analyze the time-frequency decomposition of PD signals.
2. The characteristics of the PD signal of the HHVPTL are represented by a feature vector that is used as the input of the improved BPNN algorithm that pinpoints PD patterns.
3. An IDBO-BPNN algorithm is proposed to better classify the PDs.

# FEATURE ANALYSIS AND EXTRACTION OF THE PD SIGNAL FOR HYBRID HIGH-VOLTAGE POWER TRANSMISSION LINES

## Feature analysis of PD signal

Considering the special attributions of the PD signal, the article proposes the utilization of wavelet packet decomposition (WPD) to perform time-frequency analysis on the PD signal of HHVPTLs and then analyze the features of the PD signal. When WPD is applied to analyze PD signals' multi-resolution, the Hilbert space $L^2(\mathrm{R}) = \underset{j \in z}{\oplus} W_j$ is employed to define.

Then, $\mathrm{L}^2(\mathrm{R})$ decomposition is performed based on distinct pairs of scale factors $j$ to obtain the orthogonal sum of wavelet subspace $W_j(j \in z)$ of wavelet function $\Psi(t)$. The frequency resolution can be improved by binary frequency division of the wavelet subspace (*Huang et al., 2023*; *Han et al., 2022*). The new subspace $\Omega_j^n$ is employed to characterize the scale subspace $\Gamma_j$ and the wavelet subspace $\Lambda_j$ in a uniform way. Equation (1) presents this setting.

$$\begin{cases} \Omega_j^0 = \Gamma_j \\ \Omega_j^1 = \Lambda_j. \end{cases} \tag{1}$$

Then, the decomposition of the new subspace $\Omega_j^n$ can be employed to unify the $\Gamma_{j+1} = \Gamma_j \oplus \Lambda_j$ obtained by the orthogonal decomposition of $L^2(R)$ defined by

$$\Omega_{j+1}^0 = \Omega_j^0 \oplus \Omega_j^1. \tag{2}$$

The subspace $U_j^n$ is set to be the wavelet subspace of the function $u_n(t)$ and the subspace $\Omega_j^{2n}$ is set to be the wavelet subspace of the function $u_{2n}(t)$. Then, the dual scale equation is constructed and defined (*Zhang et al., 2019*) by

$$\begin{cases} u_{2n}(t) = \sqrt{2} \sum_{k \in z} A(k) \cdot u_n(2t - k) \\ u_{2n+1}(t) = \sqrt{2} \sum_{k \in z} B(k) \cdot u_n(2t - k) \end{cases}. \tag{3}$$

where $A(k)$ and $B(k)$ represents the low-pass filter and high-pass filter coefficients, respectively, and are orthogonal to each other where Eq. (3) is transformed and rewritten by

$$\begin{cases} u_0(t) = \sqrt{2} \sum_{k \in z} A(k) \cdot u_0(2t - k) \\ u_1(t) = \sqrt{2} \sum_{k \in z} B(k) \cdot u_0(2t - k) \end{cases}. \tag{4}$$

Equation (4) presents functions that are reduced to obtain the corresponding scale function $\varphi(t)$ and the corresponding wavelet function $\Psi(t)$, respectively in the multi-resolution analysis. The dual scale equation is constructed by

$$\begin{cases} \Psi(t) = \sqrt{2} \sum_{k \in z} A(k) \cdot \Psi(2t - k) \\ \varphi(t) = \sqrt{2} \sum_{k \in z} B(k) \cdot \Psi(2t - k) \end{cases}. \tag{5}$$

By the equivalence of Eqs. (3) and (4), the condition of $n \in Z$ is deduced. Then, Eq. (2) can be rewritten by

$$\Omega_{j+1}^n = \Omega_j^{2n} \oplus \Omega_j^{2n+1}. \tag{6}$$

The essence of the WPD is to further decompose the signal into two parts, namely, low frequency and high frequency, respectively. Then, $s_j^n(t) \in U_j^n$ isset as follows:

$$s_j^n(t) = \sum r_l^{j,n} u_n(2^j t - l). \tag{7}$$

By Eq. (7), $r_l^{j+1,n}$ is resolved to obtain $r_l^{j,2n}$ and $r_l^{j,2n+1}$ defined by

$$\begin{cases} r_l^{j,2n} = \sum_k \alpha_{k-2l} \cdot r_k^{j+1,n} \\ r_l^{j,2n+1} = \sum_k \beta_{k-2l} \cdot r_k^{j+1,n} \end{cases}. \tag{8}$$

Equation (8) is the WPD algorithm, which makes it possible to divide the PD signal into different WDs, which have the same bandwidth and are connected back and forth. The number of signal points contained is folded into half of the previous layer.

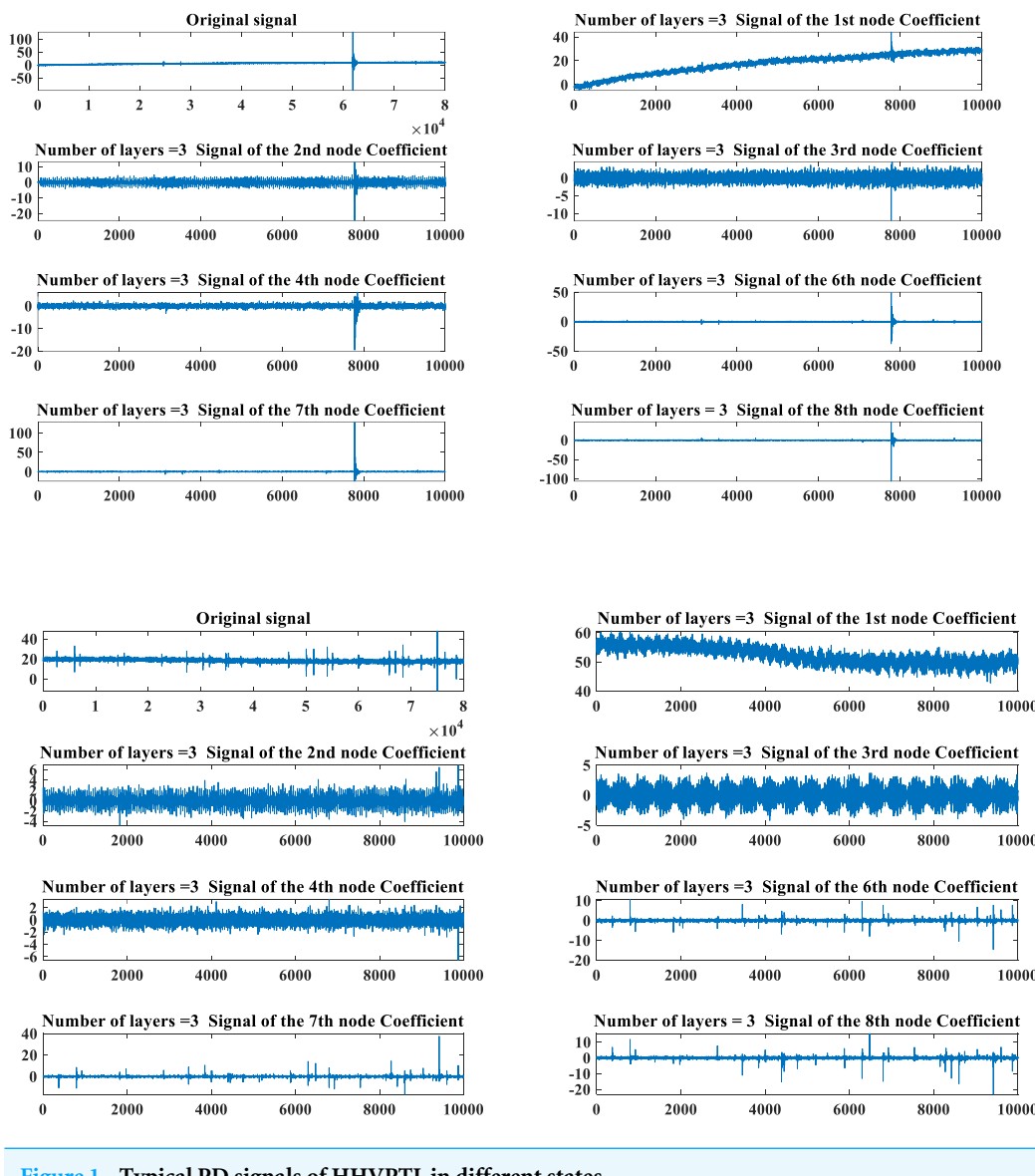

**Figure 1** Typical PD signals of HHVPTL in different states.

The PD data of HVPTLs from the Kaggle featured prediction competition is utilized. The dataset is normalized and categorized into two cases, namely, normal and faulty. The original signals of the two states and the results of the decomposition of the original signals implementing the WPD are presented in Fig. 1.

Figure 1 depicts that the PD signal composition of HHVPTLs is extremely complex, and it is difficult to guarantee higher accuracy by simply running the diagnosis manually. So, the extraction of the features of the PD signal is required to run a further diagnosis.

## Feature extraction of PD signal

Figure 1 depicts that the PD signal energy of HHVTPLs is mainly concentrated in the fundamental frequency ranging from 0~10,000 Hz. To swiftly grab the PD signal features

**Table 1  Main feature index.**

| Code | Feature index | Formula | Code | Feature index | Formula |
|------|--------------|---------|------|--------------|---------|
| $V_1$ | Main frequency energy share | $V_1 = \|x_{1,n}^2\| / \sum_{i=1}^{7}\sum_{n}^{N}\|x_{i,n}^2\|$ | $V_7$ | Minimum value | $x_{\min} = \min(x_n)$ |
| $V_2$ | Average value | $x = \sum_{N}^{n=1} x(n)/N$ | $V_8$ | Pulse Index | $I = \frac{x_p}{x}$ |
| $V_3$ | Standard deviation | $\sigma_x = \sum_{N}^{n=1}[x(n)-x]^2/(N-1)$ | $V_9$ | Skewness | $S = \sum_{N}^{n=1}[x(n)-x]^3/(N-1)\sigma_x^3$ |
| $V_4$ | Root Mean Square | $x_{\mathrm{rms}} = \sqrt{\sum_{N}^{n=1} x^2(n)/N}$ | $V_{10}$ | Cliffness | $K = \sum_{N}^{n=1}[x(n)-x]^4/(N-1)\sigma_x^4$ |
| $V_5$ | Peak value | $x_p = \max\|x(n)\|$ | $V_{11}$ | Spectral mean value | $F_{12} = \sum_{K}^{k=1} s(k)/K$ |
| $V_6$ | Maximum value | $x_{\max} = \max(x_n)$ | $V_{12}$ | Frequency center of gravity | $F_{16} = \sum_{K}^{k=1} f_k \cdot s(k) / \sum_{K}^{k=1} s(k)$ |

of HHVPTLs, the article adopts the main frequency energy share of the PD signal obtained by the WPD as the core feature index. It adopts the assessment index of the conventional mechanical signal in Table 1 as the main attribute index of the PD signal to jointly establish the feature vector of the PD signal of HHVPTLs.

Since the frequency of the PD signal is primarily concentrated in the first decomposition node, the expression for the energy share of the main frequency $V_1$ is defined by

$$V_1 = \frac{\left|x_{1,n}^2\right|}{\sum_{i=1}^{7}\sum_{n}^{N}\left|x_{i,n}^2\right|} \tag{9}$$

where $E_0$, $N$ and $x_{i,n}$ denote the fundamental frequency energy, the length of the signal data sample, and the amplitude of each discrete point after the signal was reconstructed.

The final attribute vector of the PD signal of HHVPTLs that can be obtained consists of 12 total features denoted by $V_1$ through $V_{12}$.

# IDBO-BPNN-BASED PATTERN RECOGNITION MODEL FOR LOCAL DISCHARGE SIGNALS

## BPNN

The essence of BPNN processes error terms between the actual output value and the predicted value by implementing the gradient descent algorithm. Namely, the derivative of the errors is resolved to tune the network weights and the threshold and the correction along the negative gradient of the derivative is continuously updated until the output error meets the preset requirements. When compared with the conventional algorithms, the advantage of the BPNN is that a deterministic mathematical model is no longer needed. The training stage is embedded in the model, and the corrected network can be obtained by tuning the parameters repeatedly (*Zhan et al., 2022*; *Ou et al., 2022*). The structure of BPNN is shown in Fig. 2.

The input layer in Fig. 2 contains n neurons that define the input vector, the hidden layer contains l neurons, the hidden layer contains $m$ neurons, and the output vector is defined as the last layer of the network. The link weights between the input layer and the

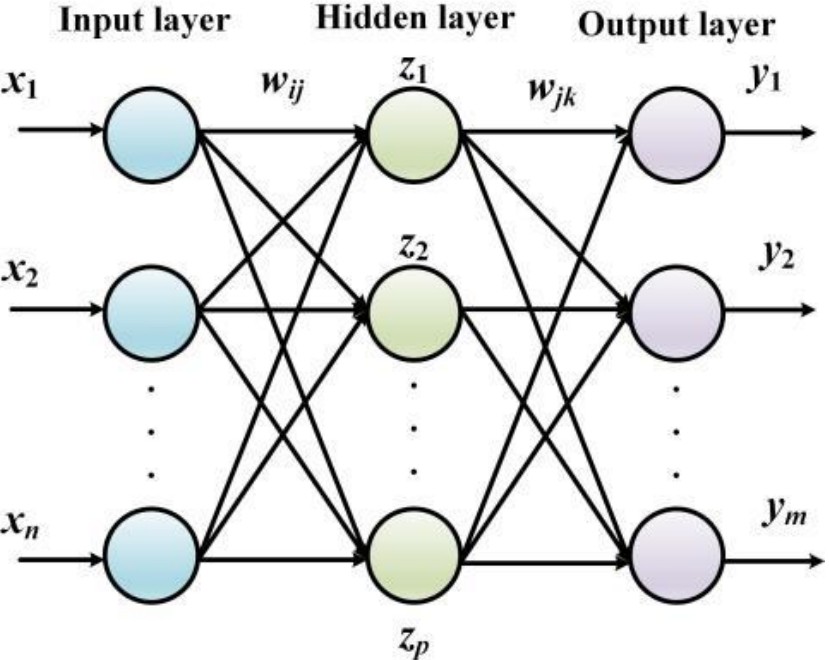

**Figure 2** **The structure of BPNN.**

hidden layer are defined as $w_{ij}$, the link weights between the hidden layer and the output layer are defined as $w_{jk}$, the threshold value of the neurons contained in the hidden layer is set to $\alpha_j (j = 1, 2, \cdots, l)$, and the threshold value of the neurons contained in the output layer is set to $\beta_k (j = 1, 2, \cdots, l)$.

In the implementation of the BPNN, the training stage first determines network weights, and thresholds are continuously corrected during the training stage to transform the network into a prediction function. The steps are presented below.

Initialize the model parameters. The number of neurons in each layer is set according to the input and output vectors of the model, the link weights, the threshold of neurons in the hidden layer, and the threshold of neurons in the output layer are initialized, and the learning rate and the corresponding excitation function of the model are set.

Calculation of hidden layer output. The output vector $H$ of the hidden layer is calculated based on the input vector of the model, the link weights between the input layer and the hidden layer, and the threshold of the hidden layer.

$$H_j = Function(\sum_{i=1}^{n} w_{ij}x_i - \alpha_j), j = 1, 2 \cdots, l. \tag{10}$$

Equation (10) represents the excitation function of the hidden layer.

Calculation of output layer output. Similar to , the network output vector $O$ is calculated by employing the output of the hidden layer, the link weights of both the hidden layer and the output layer between each other, and the threshold value of the output layer.

Equation (11) presents the output layer.

$$O_k = \sum_{j=1}^{l} H_j w_{jk} - \beta_k, k = 1, 2 \cdots, m. \tag{11}$$

Calculation of network error. The expected error $E_k$ is obtained by calculating the difference between the expected output $O_k$ of the model and the desired output $Y_k$. Equation (12) presents the error term.

$$E_k = Y_k - O_k, k = 1, 2, \cdots, m. \tag{12}$$

Correction of network weights. The link weights of the network are corrected according to the network error.

$$w'_{ij} = w_{ij} + \xi H_j (1 - H_j) x_i \sum_{k=1}^{m} w_{jk} E_k \tag{13}$$

$$w'_{jk} = w_{jk} + \xi H_j E_k \tag{14}$$

where $\xi$ denotes the learning rate, $w'_{ij}$ and $w_{ij}$ denote the updated weight at the next iteration, $w'_{jk}$ and $w_{jk}$ represent the updated weight at the next iteration.

Correction of network threshold. The network threshold is corrected according to the network error, where $\alpha'_i$ and $\beta'_k$ denote the threshold parameters to be updated at the next iteration.

$$\alpha'_i = \alpha_i + \xi H_j (1 - H_j) x_i \sum_{k=1}^{m} w_{jk} E_k \tag{15}$$

$$\beta'_k = \beta_k + E_k. \tag{16}$$

Determine whether the output meets the termination condition and terminate the iteration if the preset score is reached, otherwise, return to .

### Improvement of the BPNN based on orthogonal dyadic learning dung beetle optimization

Since the actual operation of the BPNN is mainly found by the network parameters, such as the number of nodes and connection weights, the final scores of the network parameters will have an impact on the BPNN's results. To ensure that the BPNN has a good search performance and avoids falling into local optimums, an optimization algorithm needs to be introduced to optimize network parameters.

The dung beetle optimization (DBO) is the recently developed algorithm working on population intelligence optimization proposed in 2022. It is mainly derived from the habits of dung beetle life and has a robust search capability and high convergence efficiency. Thus, the DBO algorithm is employed to optimize BPNN in the manuscript.

DBO employs tentacle navigation when rolling to ensure that the dung ball keeps a straight line forward during the rolling process, and this behavior is needed in the simulation to permit the dung beetle to advance in the search space in the set direction and assume that light intensity affects the selection of dung beetle's forward path. Then, the position of the dung beetle during the forward process can be defined by

$$x_i(t+1) = x_i(t) + \lambda k x_i(t-1) + \mu \Delta x \tag{17}$$

$$\Delta x = |x_i(t-1) - x_{worst}| \tag{18}$$

where $t$ characterizes the number of iterations so far, $x_i(t)$ characterizes the position of the $i$-th dung beetle at the $t$-th iteration, and $k \in (0, 0.2)$ characterizes the deflection factor, which is usually set to a constant value. $\lambda$ taking either $-1$ or $1$, and $\mu$ denotes a constant value in (0-1), $x_{worst}$ characterizes the local worst position, $\Delta x$ representing mainly used regulation for light intensity.

When a dung beetle encounters an obstacle and cannot move forward, it needs to re-roll to reorient itself to develop a new route. To model the rolling orientation behavior, the new direction is resolved by utilizing the tangent function defined by.

$$x_i(t+1) = x_i(t) + \tan\theta |x_i(t) - x_i(t-1)|. \tag{19}$$

In Eq. (19), $\theta \in [0, \pi]$ characterizes the angle of deflection and $x_i(t) - x_i(t+1)$ represents the difference between the anterior and posterior positions of the $i$-th dung beetle at distinct iteration cycles.

In addition, female, juvenile, and thieving dung beetles are defined in the population with position expressions as follows:

$$x_{fdb\_i}(t+1) = x_{best} + \varpi_1 \left(x_{fdb\_i}(t) - lb^*\right) + \varpi_2 \left(x_{fdb\_i}(t) - ub^*\right) \tag{20}$$

$$x_{jdb\_i}(t+1) = x_{best} + \sigma_1 \left(x_{jdb\_i}(t) - lb^c\right) + \sigma_2 \left(x_{jdb\_i}(t) - lb^c\right) \tag{21}$$

$$x_{tdb\_i}(t+1) = x_{bf} + \eta \cdot \xi \cdot \begin{pmatrix} |x_{tdb\_i}(t) - x_{best}| \\ + |x_{tdb\_i}(t) - x_{bf}| \end{pmatrix}. \tag{22}$$

To enhance the plurality of the dung beetle population, orthogonal contrastive learning is implemented to enhance it. Firstly, the orthogonal variation operator is introduced to perturb the optimal position in the population so that the joiner of the stealer masters from the variation solution of the optimal position of the finder, which can increase the population diversity with the ability to jump out of the local optimum, then Eq. (22) can be rewritten as

$$x_{tdb\_i}(t+1) = x_{bf} + \hbar \cdot \eta \cdot \xi \cdot \begin{pmatrix} |x_{tdb\_i}(t) - x_{best}| \\ + |x_{tdb\_i}(t) - x_{bf}| \end{pmatrix}. \tag{23}$$

In Eq. (17), $\hbar = \begin{cases} -1, & r < 0.5 \\ 1, & else \end{cases}$ r¡0 is defined.

Contrastive learning is a commonly implemented strategy to jump out of the position found as the local optimal solution. In the detection of the DBO, the dung beetles in the optimal position move toward the worst solution, and the dung beetles in other positions move toward the optimal score. Although searching for the worst solution position can avoid falling into the local optimum to some extent, this is not conducive to the convergence of the population. In contrast, contrastive learning not only helps individuals to escape from the current position quickly, but also the contrastive position is more likely to have a better fitness score than the current worst position, so the contrastive learning strategy with these beneficial properties is introduced into the position update formula for rolling directional advance, and Eq. (19) is improved as follows:

$$x_i(t+1) = \begin{cases} lb + ub - \delta x_i(t), f[x_i(t)] \leq f[x_i(t-1)] \\ x_i(t) + \delta \cdot \tan\theta \, |x_i(t) - x_i(t-1)|, else \end{cases}. \tag{24}$$

In Eq. (24): $\delta$ and $\beta$ represent random numbers following N(0,1), *lb* and *ub* denote the lower and upper limits of the merit search, respectively.

Then, the overall optimization process of the IDBO-BPNN is presented as follows:

Step 1: Collect the local discharge data of HVPTLs, divide the training set and test set according to phase and state, and input the training set into the model for learning.

Step 2: Initialization of the BPNN network parameters, DBO population, and algorithm parameters.

Step 3: Solve all individual fitness scores based on DBO according to the objective function.

Step 4: Update the dung beetle positions and determine if they are out of bounds by utilizing the orthogonal opposition learning algorithm.

Step 5: Update the optimal position of the dung beetle and the fitness score.

Step 6: Repeat the above steps until the iteration limit is reached; if not, then skip to step 2; otherwise, output the global optimal solution and fitness score to the BPNN.

Step 7: Output the local release pattern recognition result of HHVPTLs.

# EXPERIMENTAL TEST

## Data source and experimental environment

The data sources are extracted from the Kaggle featured prediction competition. The data are initialized to obtain 8,188 sets of normal PD signals and 8,188 sets of faulty PD signals, each with a measurement time of 20ms, containing 800,000 points, and a line operating frequency of 50 Hz. Each group of signals in the data set will experience a complete three-phase AC power supply operating cycle, and all three phases are measured simultaneously.

A method to recognize the local discharge pattern of the HVVTPL is proposed in the article. Tests were conducted on a computer whose experimental environment is shown in Table 2.

**Table 2  Experimental environment.**

| Type | Name | Content |
|---|---|---|
| Software environment | Operating system | Windows 10 64-bits |
|  | Test environment | Matlab 2018B |
| Hardware environment | CPU | Intel(R) Core(TM) i7-9750H CPU @ 2.60 GHz 2.59 GHz |
|  | GPU | NVIDIA GeForce GTX 1650 |
|  | Memory | 32 GB 2400MHz DDR4 |

## Experimental results

To test the accuracy and efficiency of the proposed IDBO-BPNN based on the data set composed of 12,600 subsets of data in the form of the PD signal of HVPTLs, normal and faulty data constitute half of the total number of the dataset. The training and test sets are divided according to the ratio of 7:2. The dataset is utilized as the input to the IDBO-BPNN to train the network. Then, the test set is employed to test the trained IDBO-BPNN network to check whether the constructed network recognizes the fault state of PD signals and calculates the accuracy. Then, the outcomes are compared with the computations of the backpropagation neural network (BPNN), support vector machine (SVM), Elman neural network ELM, deep belief network (DBN), particle swarm optimization-backpropagation neural network (PSO-BPNN) gray wolf optimization- backpropagation neural network (GWO-BPNN), genetic algorithm- backpropagation neural network (GA-BPNN), and dung beetle optimization-backpropagation neural network (DBO-BPNN) (*Wu et al., 2021*; *Jiang et al., 2021*; *Dan et al., 2021*). The obtained outcomes are shown in Fig. 3. While 0 indicates a normal PD signal, 1 designates a faulty PD signal.

Figure 3 depicts that the unoptimized BPNN has a relatively low accuracy of 71.8333% in the recognition of PD signals due to the limitations of its structure and the incapability of the iterative algorithm. In contrast, the recognition accuracy of machine learning algorithms such as SVM and ELM, the accuracy reaches only 76.5% and 74.8333%, respectively. In addition, the recognition accuracy of the DBN reaches 82.1667%. Due to their limitations, the above methods are not efficient in extracting features of PD signals, which leads to unsatisfactory recognition accuracy as well. After implementing the standard optimization algorithms such as particle swarm optimization (PSO), gray wolf optimization (GWO), and genetic algorithm (GA) to optimize the BPNN, the recognition accuracy of the obtained PD signal reaches 84.3333%, 86.5%, and 89.6667%, respectively, which is an important enhancement when compared with the unoptimized BPNN. The improvement of the BPNN based on the optimization algorithms is a better approach to attain higher accuracy.

After implementing the DBO to optimize the BPNN, the recognition accuracy of the DBO-BPNN achieves 90.1667%, which is higher when compared with all implemented algorithms, which indicates that the DBO algorithm optimizes the BPNN better when compared with other optimization methods. However, an accuracy error close to 10% is hardly acceptable in engineering implementations. Therefore, the recognition accuracy of the IDBO-BPNN obtained by implementing the DBO is improved with the orthogonal opposition learning algorithm to deal with this issue. Thus, the IDO-BPNN can achieve

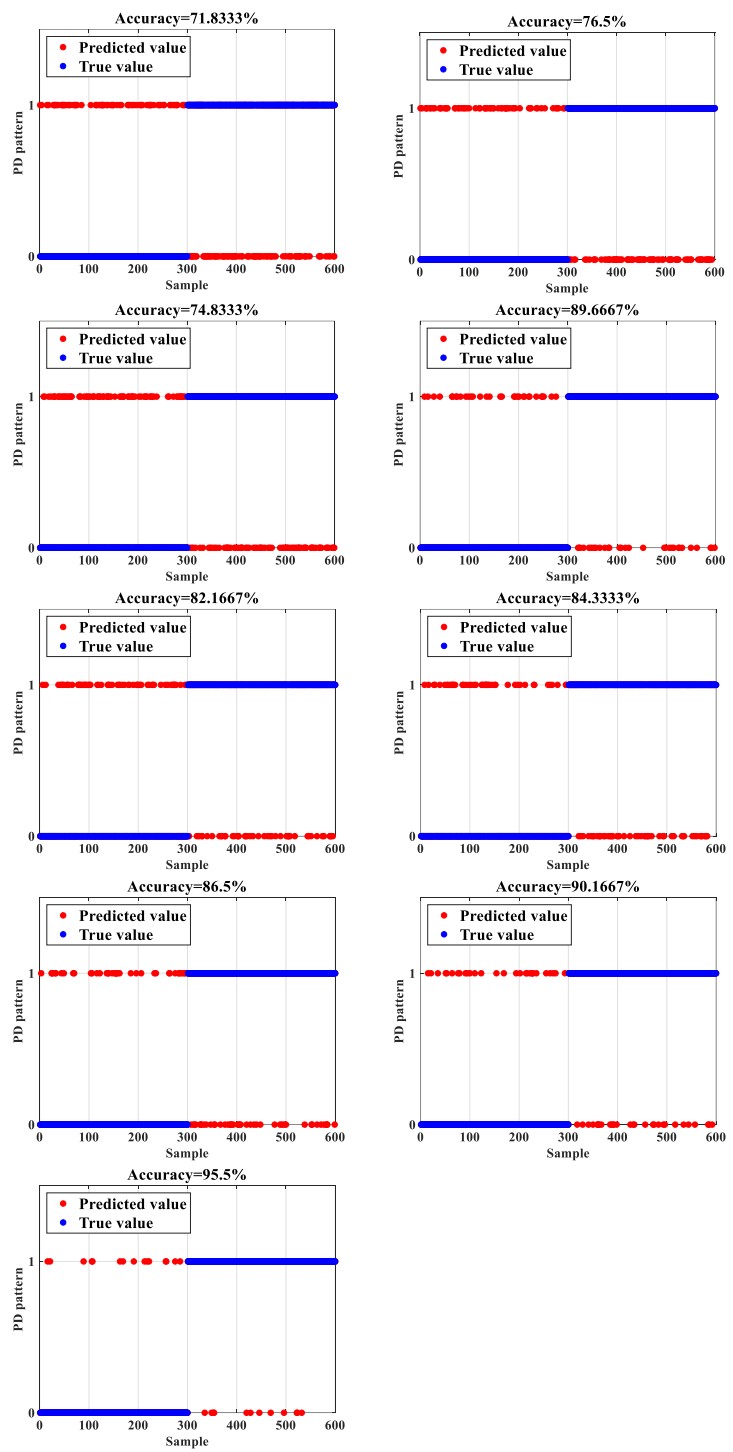

**Figure 3    Recognition results (A) BPNN (B) SVM (C) ELM (D) DBN (E) PSO-BPNN (F) GWO-BPNN (G) GA-BPNN (H) DBO-BPNN (I) IDBO-BPNN.**

**Table 3** Operational efficiency comparison results.

| Algorithm | Time (s)(Average of 10 operation cycles) |
| --- | --- |
| BPNN | 36.4 |
| SVM | 48.6 |
| ELM | 30.2 |
| DBN | 130.7 |
| PSO-BPNN | 81.8 |
| GWO-BPNN | 97.9 |
| GA-BPNN | 101.6 |
| DBO-BPNN | 67.3 |
| IDBO-BPNN | 62.6 |

95.5%, which is 5.3333% higher than the recognition accuracy of the DBO-BPNN and far more than other recognition algorithms. Therefore, PD signal patterns can be recognized more accurately and present practical performance.

## DISCUSSION

To further present the advantages of the IDBO-BPNN, the analysis of the computational efficiency of distinct algorithms is carried out. The mean computation time of distinct algorithms after 10 cycles is calculated and compared and the outcomes are summarized in Table 3.

Table 3 shows that the BPNN, as well as other machine learning algorithms, are relatively fast, basically not exceeding 50s, which is much lower than the 130.7s required for running the DBN since they are composed of simpler internal structures when compared to that of deep networks. When the optimization algorithm is implemented to improve the BPNN, the computation time increases significantly along with the improvement of recognition accuracy. Then the DBO can find the optimal solution of the BPNN faster than other optimization algorithms, so the time is significantly decreased. However, it is still higher than when the BPNN is not optimized. On the other hand, the balance between accuracy and efficiency has been achieved as much as possible. The IDBO, however, can help the BPNN reach the global optimum faster and improve the convergence efficiency of the network by implementing the orthogonal-contrastive learning algorithm, which is a better way to achieve a "win-win" situation in accuracy and efficiency.

In summary, the suggested IDBO-BPNN can efficiently and accurately detect the PD signal of HHVTPLs, which is suitable for the actual implementations and is a practical digital technology that can help improve the quality and efficiency of operation and inspection of power enterprises.

## CONCLUSION

In the article, a parallel identification method of PD patterns in HHVPTLs is suggested based on wavelet analysis and IDBO-BPNN, resolving the energy share of the main frequency of PD signal by the WPD algorithm. The feature vector together with the time domain index

and frequency domain index of the PD signal is constructed. The BPNN is improved by implementing the DBO algorithm based on orthogonal contrastive learning. Then, the suggested IDBO-BPNN can efficiently and accurately recognize PD patterns and diagnose PD faults. The experimental results show that the suggested IDBO-BPNN can balance accuracy and efficiency concurrently and has the potential to be broadly implemented in practical implementations, which can help the digital transformation of the smart grid.

The current research has some limitations as follows: 1. Only one data set is used to conclude even though the implemented dataset is a benchmark dataset. 2. The proposed method is compared with several robust machine learning methods, however, other deep learning models can be used to supplement more comprehensive outcomes. 3. The current level of accuracy is very high based on the DBO heuristic optimization algorithm. However, a comprehensive check could be done with similar algorithms to attain better results.

### Funding

The authors received no funding for this work.

### Competing Interests
Yongye Wu is employed by Chengdu Power Supply Company, State Grid Sichuan Electric Power Company. Chengdu, China. The company has no objection to publish this study.

### Author Contributions

- Yang Yang conceived and designed the experiments, analyzed the data, prepared figures and/or tables, authored or reviewed drafts of the article, and approved the final draft.
- Yongye Wu conceived and designed the experiments, performed the experiments, analyzed the data, authored or reviewed drafts of the article, and approved the final draft.
- Yifei Gao conceived and designed the experiments, analyzed the data, performed the computation work, prepared figures and/or tables, authored or reviewed drafts of the article, and approved the final draft.
- Yixuan Huang conceived and designed the experiments, performed the computation work, authored or reviewed drafts of the article, and approved the final draft.
- Shukun Liu conceived and designed the experiments, performed the experiments, performed the computation work, prepared figures and/or tables, authored or reviewed drafts of the article, and approved the final draft.
- Yuanshi Wang conceived and designed the experiments, performed the experiments, performed the computation work, prepared figures and/or tables, authored or reviewed drafts of the article, and approved the final draft.

### Data Availability
The code is available in the Supplementary File.

The data is available at Kaggle and figshare:

-https://www.kaggle.com/c/vsb-power-line-fault-detection/data.

- Klein, Lukáš; Fulneček, Jan; Seidl, David; Mišák, Stanislav; Dvorský, Jiří; Piecha, Marian (2023). A data set of Signals from an Antenna for Detection of Partial Discharges in Overhead Insulated Power Line. figshare. Collection. https://doi.org/10.6084/m9.figshare.c.6628553.v1.

## Supplemental Information

Supplemental information for this article can be found online at http://dx.doi.org/10.7717/peerj-cs.2045#supplemental-information.

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
