# Peer review of "Detection of partial discharge patterns in hybrid high voltage power transmission lines based on parallel recognition method"

_PeerJ Computer Science, doi:10.7717/peerj-cs.2045_

## Round 0.1 · original submission · Major Revisions

The reviewers have suggested a major revision to the manuscript. Authors are required to address all the comments and suggestions of the reviewers and resubmit a revision. I also have a few suggestions:

1) The term "big data" in the title seems vague and redundant as it has not been discussed how considered data are big data. How big data were processed?

2) The abstract needs modifications, as some of the abbreviations were directly used without explanation, such as BP, IDBO, IWSO, etc. Further, achieved accuracy needs to be reported in the abstract.

3) The entire manuscript needs thorough proofreading to correct typos, grammar and explanation of abbreviations during their first occurrence within the text. For instance, line 62: F-P, line no. 65, UHF, PRPD, PRPS, line 67: PSA, line 68: CNN, and many more.

4) The contribution statement is missing in the manuscript. The novelty and contributions of the manuscript must be mentioned, especially at the end of section 1.

5) Line 224-254, the steps of the IDBO-BPNN methods may be written in pseudocode form for better understanding.

6) Fig. 3: "Recognition results" are very hard to follow. Figures may be converted to more interpretable plots and each plot must be labeled with the name of the method.

7) The highest reported prediction accuracy is 95.5% which belongs to IDBO-BPNN. The authors must compare it with other published methods. As the data taken from Kaggle, there are various published methods available for comparison.

8) Authors need to mention the assumptions and limitations in the conclusion section.

**Language Note:** The Academic Editor has identified that the English language must be improved. PeerJ can provide language editing services - please contact us at [email protected] for pricing (be sure to provide your manuscript number and title). Alternatively, you should make your own arrangements to improve the language quality and provide details in your response letter. – PeerJ Staff

Reviewer 1 ·

Basic reporting

The manuscript lacks proper English, and it is highly advised to complete the proofreading with professional services / fluent English support speakers.
The Literature Review should introduce the State of Art Section in an organized way.
The figures submitted by the authors are unclear and per the Journal Guidelines. The figures should be modified for clarity of vision and understanding into vector graphics.
Several terms in the Formal results do not include the definitions of the terms used in the equations and theorems.
It is highly recommended that these formulae be updated with proper terms and meanings.

Experimental design

Detection of partial discharge patterns in hybrid high voltage power transmission lines based on big data-based parallel recognition method, feature extraction, and adaptive weighting is proposed to address the challenges of diverse types in this study.

It is recommended that the authors include the definitions and proper mathematical explanations of the terms used in the study, such as Equations 1, 2, 3, 4, 5, 6, 7, 8, etc.

The experimental setup needs more elaboration of the terms used in the article for the readers to understand the paper more clearly.

The methodology must be explained in a more detailed way to help the reader understand the parallel recognition approach in the given study.

Various terms are not explained for the readers to make it understandable. It is recommended that the same is explained for all the terms used in the manuscript.

Validity of the findings

The datasets used for the PD signal of HVPTL are obsolete and legacy data. It is highly advised to the authors so that they can train the proposed study using some of the latest research data.

Some enhanced data sets with better real-time inputs can be used for the complete training of the algorithm to prove the accuracy of the algorithm proposed.

Similar kind of methodologies are observed in the manuscripts:

https://patents.google.com/patent/WO2004034070A1/it
https://ieeexplore.ieee.org/abstract/document/8341662
https://stax.strath.ac.uk/downloads/5712m6923
https://www.preprints.org/manuscript/202306.0942/v1
https://www.mdpi.com/2073-8994/14/11/2464
https://ietresearch.onlinelibrary.wiley.com/doi/full/10.1049/smt2.12137
https://ieeexplore.ieee.org/abstract/document/6378733

The novelty of research by the authors should be proved by a comparative study of all the State-of-the-art methods.

Additional comments

Several new citations and references from recent inventions should be included in the manuscript.
Kindly ensure the Table Citations and Figure Referencing appear in an ascending sequence in the manuscript.
All the figures under the results section must explain the findings and then compare them with the previous models used for similar kinds of study as per the literature review.
The paper contains potential, but with the aforesaid changes, it can be more beneficial and acceptable.
Authors must complete the reviews and modify the findings per the above sections.

Reviewer 2 ·

Basic reporting

Improve the quality of Figure 1 and 3 with clear visibility and also mention units for X and Y axis.
Refer the following papers to improve the Literature references,
1) Detection Method of Partial Discharge on Transformer and Gas-Insulated Switchgear: A Review
2) Detecting power systems failure based on fuzzy rule in power grid
3)Review on Partial Discharge Diagnostic Techniques for High Voltage Equipment in Power Systems
4)Student Psychology based optimized routing algorithm for big data clustering in IoT with MapReduce framework
Keywords should be expanded words.
Figure 2 shows The structure of BPNN. Need to relate with proposed work.

Experimental design

EXPERMENTAL Setup with number of resources required- State Clearly

Validity of the findings

Add the limitation and Objective of the proposed work in Introduction Section
CONCLUSION section highlights with result values.
future direction not given

Additional comments

nil

---

## Round 0.2 · Minor Revisions

Reviewers recommended few more changes to the manuscript. You are required to address these issues and resubmit a revision.

Reviewer 1 ·

Basic reporting

The authors have revised the English and language as per the directions.
The authors must process the graphics according to the journal guidelines regarding clarity.

Experimental design

The authors have explained the methodology and terms needed in the new revision.

Validity of the findings

The discussion of results is presented well in the manuscript.

Additional comments

However, the authors post all the requested information, and the quality of Figures 1, 2 and 3 is not relevant to the journal guidelines.
The authors must process the graphics according to the journal guidelines in terms of clarity.

Reviewer 3 ·

Basic reporting

- This manuscript describes IDBO-BPNN, which is implemented to investigate the local
discharge data of a hybrid high-voltage power transmission lines in the Kaggle Featured Prediction Competition.

- This paper needs improvement in the present form.

- Highlight main contributions in 3-4 points in the Introduction section.

- Add a paragraph in the last of Introduction section to highlight rest of sub-sections.

- Add a literature review section after Introduction section.

- Highlight why your study is better than previous works.

- Figure for the architecture of the IDBO-BPNN is missing. Figure-2 depicts only normal Back propagation flow.

- Add a section to describe the proposed DBO-BPNN in detail.

Experimental design

- Authors considered only one dataset taken from Kaggle Featured Prediction Competition. Try to run the experiment on other dataset as well.

- Describe and cite all comparing methods given in Table-3.

- Add Ablation study in the experimental section.

- Add a comparative analysis sub-section

Validity of the findings

- Add a separate sub-section to highlight limitations and future research scope of this work.

- Add a Discussion section in the experimental section.

Additional comments

Major revision is needed.

---

## Round 0.3 · accepted · Accept

I am pleased to inform you that your paper has been accepted for publication in PeerJ Computer Science. Your manuscript has undergone rigorous peer review, and I am delighted to say that it has been met with high praise from our reviewers and editorial team. On behalf of the editorial board, I extend our warmest congratulations to you.

Reviewer 1 ·

Basic reporting

All comments are answered by the authors.

Experimental design

All comments are answered by the authors.

Validity of the findings

All comments are answered by the authors.

Additional comments

The authors have processed the graphics according to the journal guidelines regarding clarity.

Reviewer 3 ·

Basic reporting

Authors have addressed all my comments.

Experimental design

Authors have addressed all my comments from experimental section perspectives.

Validity of the findings

All comments are properly addressed in the present form.

Additional comments

Accepted in the present form.